# Association of Upper Lip Morphology Characteristics with Sagittal and Vertical Skeletal Patterns: A Cross Sectional Study

**DOI:** 10.3390/diagnostics11091713

**Published:** 2021-09-18

**Authors:** Xinyu Yan, Xiaoqi Zhang, Yiyin Chen, Hu Long, Wenli Lai

**Affiliations:** National Clinical Research Center for Oral Diseases, State Key Laboratory of Oral Diseases, Department of Orthodontics, West China Hospital of Stomatology, Sichuan University, No.14, Section 3, Ren Min South Road, Chengdu 610041, China; xinyuyan100@163.com (X.Y.); zhangxiaoqi67@163.com (X.Z.); hxkqchenyiyin@163.com (Y.C.); apprehendall@hotmail.com (H.L.)

**Keywords:** upper lip, soft tissue, skeletal pattern, cephalometrics, multivariate regression

## Abstract

Background: Upper lip morphology is essential in diagnosis and treatment of orthodontics and orthognathic surgery. This study is aimed to evaluate the association between upper lip characteristics (ULCs) and skeletal patterns (SPs). Methods: 2079 patients were involved and grouped by sagittal and vertical. Class I, II, and III were identified by ANB angle, while normodivergent, hyperdivergent, and hypodivergent were identified by Facial Height Index and Sum of Angles. ULCs were evaluated by superior sulcus depth, nasolabial angle, upper lip length, basic upper lip thickness, and upper lip thickness. Confounders including demography, malocclusion, upper incisors, and upper lips were adjusted by multivariate linear regression to identify the association between ULCs and SPs. Group differences were evaluated with analysis of variance and Chi-square test. Results: The mean value of ULCs and prevalence of SPs were explored in the Western China population. ULCs were significantly different in various sagittal, vertical, and combined SPs. Superior sulcus depth was negatively related to Class II, and positively related to Class III and the hypodivergent pattern after adjusted by confounders. Conclusions: ULCs significantly varied among different SPs, while only superior sulcus depth was independently associated with SPs, indicating superior sulcus depth is the only ULC that might be significantly corrected by intervention of skeletal growth.

## 1. Introduction

Profile esthetics is generally becoming one of the major objectives of patients seeking orthodontic or orthognathic therapies nowadays. Therefore, it is of great significance to establish individualized standard of profile beauty and distinguish its underlying correlated factors.

Facial profile is considered dependent on dentoskeletal tissue and overlying soft tissue. Skeletal patterns are reflections of the relative position of maxilla and mandible, which can be evaluated in the sagittal and vertical dimension, respectively. The sagittal skeletal pattern indicates the anteroposterior displacement of upper and lower jaw, which is classified into three types: Class I indicates maxilla and mandible are in harmonious relative position; Class II indicates the maxilla is relatively prognathic compared with mandible; and Class III indicates more protruded mandible relative to maxilla. Vertical skeletal patterns usually result from the growth and rotation of mandible, which can be divided into hyperdivergent (increased mandibular plane angle and clockwise rotation), hypodivergent (decreased mandibular plane angle and counterclockwise rotation), and normodivergent patterns [1]. Accumulated evidence has demonstrated that both sagittal and vertical skeletal patterns greatly impact on soft tissue morphology [2,3,4]. For instance, a study on an Indonesian population found that the upper lip was generally deeper in Class III compared to Class II [5]. Similarly, it was shown that facial soft tissue thickness, especially in the chin area, varied among different vertical developmental patterns, which was smaller in hyperdivergent patterns [6,7].

Traditional opinions thought that soft tissue characteristics greatly resemble its adherent hard tissue patterns [8,9]. However, increasing evidence has indicated that multiple confounding factors involving genetics and environment (gender, race, age, etc.) increase the variability of soft tissue morphology. For example, males are found to have thicker lower third facial soft tissue than female with the same skeletal patterns [6,10]. Moreover, dental characteristics, such as crowding, occlusal relationship, and incisor position, also play an influential role. Specifically, anteroposterior upper incisor position was found greatly associated with upper lip thickness [11,12]. Hence, multiple confounding variates blur the definite relationship between soft tissue and skeletal patterns.

Cephalometric analysis is the most common diagnosis approach in orthodontic treatment due to its simplicity and reliability, which helps orthodontists to evaluate sagittal and vertical jaw relationship, soft tissue characteristics, dental malocclusion, and growth tendency through 2D images [13,14]. A series of lateral cephalometric analysis methods have been developed so far, with standardized values established in certain populations with a norm skeletal pattern and occlusion as well as esthetic profile [15,16]. Nevertheless, considering the unneglectable impact of different skeletal patterns on soft tissue characteristics, it is essential to explore the precise relationship between these two using very strict statistics to control abovementioned confounding factors. Thus, it facilitates orthodontists making more individualized treatment plans for patients with different skeletal patterns, instead of simply following norm values for normal populations.

The lip is the pivotal feature affecting the esthetics of the lower third of face, especially as upper lip attracts the greatest attention [17]. It has been proven that vertical lip thickness is the most influential variable in smile esthetics [12]. Current studies on the associations of soft tissue and skeletal patterns mainly focused on nose and chin area, and the studies concerning the upper lip region are relatively rare, also defective in limited sample size or poor statistical methodology [18,19]. Therefore, we conducted a statistically well-designed study on Chinese population with the largest sample size to date, aimed to (1) establish the mean value of upper lip characteristics in Western China population; (2) compare upper lip characteristics in different sagittal and vertical skeletal pattern groups; and (3) explore the skeletal patterns which independently affects upper lip characteristics despite of all confounding variates. The null hypothesis of the study was that upper lip characteristics were not significantly different among skeletal pattern groups.

## 2. Materials and Methods

### 2.1. Study Population

The study was a cross-sectional study, which was reported following the Strengthening the Reporting of Observational Studies in Epidemiology (STROBE) guideline [20]. The flowchart of the investigating process is shown in Figure 1.

Patients receiving orthodontic treatment in the Department of Orthodontics, West China Hospital of Stomatology, Chengdu, Sichuan from January 2013 to December 2020 were retrospectively identified. The exclusion criteria were used as follows: (1) participants without complete baseline diagnostic information; (2) participants aged less than 12 years old; and (3) participants without permanent dentitions. All participants received a series of examinations prior to orthodontic treatment, including demographic questionnaires, radiographic inspections, plaster or digital models, and oral photographs. The study was approved by ethics committee of West China Hospital of Stomatology, and written informed consent was obtained from every adult participant and the guardian of every minor.

### 2.2. Upper Lip Characteristics and Skeletal Patterns

Lateral cephalograms were taken by the same device (Veraviewepocs, Morita, Kyoto, Japan) with the patients in centric occlusion. Cephalometric measurements before orthodontic treatment were conducted using Dolphin imaging software version 11.9.07.23 (Patterson Dental, Los Angeles, CA, USA) by the same experienced orthodontist. The cephalometric landmarks involved in this study were explained as follows:Porion (Po): the midpoint of the upper contour of the external auditory canal.Orbitale (Or): the lowest point on the inferior margin of the orbit.Nasion (N): the most anterior point on midline of frontonasal suture.Columella (Cm): The most prominent point on the borderline between lower part of the nose contour and nasal tip.Subnasale (Sn): the deepest point on the curvature between the anterior nasal spine (ANS) and the prosthion on the anterior surface of the maxilla.Subspinale (point A): the innermost point on the contour of the premaxilla between ANS and the incisor tooth.Supramental (point B): the innermost point on the contour of the mandible between the incisor tooth and the bony chin.Labrale superius (UL): the most anterior and convex point of upper lip vermilion.Stomion superius (Stms): the lowest point of the margin of upper lip vermilion.

Upper lip characteristics were described using five indices, including upper lip length, upper lip thickness, basic upper lip thickness, superior sulcus depth, and nasolabial angle. Specifically, upper lip length is the vertical distance between Sn and Stms; upper lip thickness is the distance from the labial surface of upper incisor to UL; basic upper lip thickness is the distance from the point 3 mm below point A to Sn; the superior sulcus depth is the distance from the most concave point of upper lip to the line vertical to Frankfort (FH) plane, which is a line through Or and Po; and the nasolabial angle is the angle between the line connecting Cm and Sn and the line connecting Sn and UL (Figure 2A).

Skeletal patterns were evaluated in the sagittal and vertical dimensions. In sagittal dimension, skeletal patterns were divided into Class I, II, and III according to the angle of ANB, which is the angle between the line connecting point A and N and the line connecting point B and N. Class I is defined as ANB ≥ 1° and ≤ 5°, Class II is defined as ANB > 5°, and Class III is defined as ANB < 1°, which is more suitable for Chinese population [21,22]. In vertical dimension, three skeletal patterns were distinguished by Jarabak’s ratio (FHI, Facial Height index) and Björk’s sum (SOA, Sum of Angles) [23], which indicates a normodivergent pattern when 61% ≤ FHI ≤ 65% & 300° ≤ SOA ≤ 402°, a hypodivergent pattern when FHI > 65% & SOA < 300°, and a hyperdivergent pattern when FHI < 61% & SOA > 402° (Figure 2B,C).

### 2.3. Demographics and Covariates

Demographic information, including age and gender, was acquired from questionnaires, and participants was divided into adolescent (12–18 years old), young adult (18–35 years old), and middle age (≥35 years old) according to age. Molar relationship, crowding, overbite and overjet were assessed based on models and oral photographs. Molar relationship is diagnosed with I when the mesiobuccal cusp of the upper first molar (U6) occludes with the buccal groove of lower first molar (L6), with II-1 when L6 is distal to U6 and upper incisors proclined, with II-2 when L6 is distal to U6 and upper incisors retroclined (U1-SN < 100°), with III when L6 is mesial to U6, and with IV when the molar relationship is II on one side and III on the other. Crowding is evaluated as I (<4 mm), II (4–8 mm), and III (≥8 mm). Overbite is divided as open (<0 mm), shallow (0–1 mm), normal (1–5 mm), and deep (>5 mm). Overjet is divided into cross (<0 mm), shallow (0–1 mm), normal (1–5 mm), and deep (>5 mm). Above classification was based on the references that are more suitable for the Chinese population [21,22]. Moreover, other cephalometric indices were also considered as covariates, mainly involving the relative position of the upper and lower lips (upper lip to S line (UL-SL), the upper lip to E line (UL-EL)), as well as the position and inclination of the upper incisor (U1-ANS, U1-OP, U1-PP, U1-NA, U1-SN, U1-NPo).

### 2.4. Statistical Analysis

Continuous variables were described as means (standard deviations, SDs), and categorical variables were expressed as frequencies (percentages). Demographic and baseline clinical characteristics were compared among groups with different sagittal skeletal patterns, vertical skeletal patterns, and sagittal-vertical combined skeletal patterns using analysis of variance and the Chi-square test as appropriate.

Multivariate linear regression models were used to compare five indices of upper lip characteristics in different sagittal and vertical skeletal patterns after adjusting all covariates, including age, gender, molar relationship, crowding, overbite, overjet, cephalometric indices of U1, and upper lip morphology. Further, we investigated the independent association of upper lip characteristics with skeletal patterns with adjustment of different covariates in the following models. Model 1 was adjusted for age, gender, molar relationship, upper crowding, lower crowding, overbite and overjet; model 2 was adjusted for “U1-ANS (mm)”, “U1-OP”, “U1-PP (mm)”, “U1-NA”, “U1-NA (mm)”, “U1-SN”, “U1-PP” and “U1-NPo (mm)”; model 3 was adjusted for “Nasolabial A”, “UL-EP (mm)”, “upper lip to S line”, “upper lip length (ULL) (mm)”, “basic upper lip thickness (mm)” and “upper lip thickness (mm)”; and model 4 was adjusted for all of above variables. To further explore the stratified relationship between upper lip characteristics and skeletal patterns, the skeletal patterns were classified into nine sagittal-vertical combined types, and the subgroup analysis was conducted. The multivariate linear regression results were depicted as β and 95% confidence interval (CI).

All the variables were detected by two examiners. For demographic characteristics (age and gender), they were directly exported from the hospital record system. For categorical variables (molar relationships), all authors were involved in discussion of the suitable classification if the initial results of two examiners were inconsistent. For continuous variables, intraclass correlation coefficient (ICC) was used, and all the ICC values were >0.9 with corresponding *p*-values < 0.05, indicating that the consistency of the intra-examiner agreement was very reliable. All statistical analysis was conducted using R software (version 4.0.4), and a two-sided *p* value < 0.05 was considered as statistically significant.

## 3. Results

### 3.1. Study Participant Characteristics

To obtain the whole picture of upper lip characteristics in the Western China population, 2617 patients were initially identified, and 2079 were remained as final participants after applying exclusion criteria. Among the participants included in this study, the four cephalometric indices of upper lip characteristics were normally distributed, and only upper lip thickness was not (Figure 3A). The mean and standard deviation of each characteristic was specified in Table 1. In sagittal skeletal patterns, Class I was the most prevalent with a percentage of 46.56%, Class II was the secondly prevalent (33.24%), and Class III was the least frequent (20.2%). In vertical skeletal patterns, the normdivergent pattern was the most prevalent (63.2%), orderly followed by the hypodivergent pattern (25.3%), and then the hyperdivergent pattern (11.5%). Combining sagittal and vertical skeletal patterns, it is predictable that Class I normodivergent cases were the most common (30.59%), and that the Class III hyperdivergent pattern was the rarest (1.3%) (Figure 3B).

Other clinical characteristics of the study population are presented in Table 1. Briefly, 2079 participants consisted of 623 men and 1456 women, and the majority were young adults (61.2%), with a molar relationship of I (31.5%) or II-1 (31.5%), I degree crowding (61.3% in upper arch and 63.7% in lower arch), and normal overbite (62.6%) and overjet (54.1%). Besides, the average values of other cephalometric indices on the position of U1 and upper lip were also shown, including U1-ANS, U1-OP, U1-PP, U1-NA, U1-SN, U1-NPo, and UL-EP.

### 3.2. Comparison of Upper Lip Charactristics among Different Skeletal Patterns

To obtain a comprehensive understanding on upper lip characteristics among different skeletal patterns, the comparisons among different skeletal patterns were performed after adjusting all covariates using multivariate linear regression models, considering the real-world situations and influences from various confounders (Figure 4 and Appendix A). The differences in five cephalometric indices among three sagittal skeletal patterns were statistically significant (all *p* < 0.001) (Figure 4A–E). Adjusted upper lip thickness was significantly larger in Class III (15.69 ± 2.20 mm) and smaller in Class II (14.40 ± 1.87 mm). Similarly, basic upper lip thickness was also the largest in Class III (14.74 ± 1.63 mm) and smallest in Class II (14.33 ± 1.50 mm). In terms of superior sulcus depth, Class III showed the largest value (5.45 ± 2.05 mm) while Class II showed the smallest (4.44 ± 1.92 mm). On the contrary, the largest value of upper lip length was in Class II (22.16 ± 1.85 mm) and the smallest was in Class III (20.41 ± 1.90 mm). Furthermore, the nasolabial angle was significantly smaller in Class III (89.96 ± 8.80°) and larger in Class II (99.17 ± 7.91°).

On the other hand, the differences among vertical skeletal patterns were also statistically significant for all the upper lip characteristics, except for adjusted upper lip thickness (Figure 4F–G). Both adjusted basic upper lip thickness and superior sulcus depth were significantly larger in the hypodivergent pattern (15.18 ± 1.57 mm and 5.28 ± 2.10 mm, respectively), and smaller in the hyperdivergent pattern (14.14 ± 1.62 mm and 4.34 ± 2.03 mm, respectively). Oppositely, the adjusted upper lip length and nasolabial angle were larger in the hyperdivergent pattern (22.43 ± 1.88 mm and 97.57 ± 8.45°), and smaller in the hypodivergent pattern (20.78 ± 1.95 mm and 93.05 ± 9.19°).

### 3.3. Associations between Upper Lip Characteristics and Skeletal Patterns

To make an exploration of whether skeletal patterns independently affect upper lip characteristics, the associations among different skeletal patterns were analyzed using multivariate linear regression models and reported as Table 2. Superior sulcus depth was negatively related to Class II (β = −0.195, 95% CI −0.302 to −0.087; model 4), and positively related to Class III (β = 0.253, 95% CI 0.117 to 0.389; model 4). While, in vertical dimension, superior sulcus depth was positively correlated with the hyperdivergent pattern (β = 0.185, 95% CI −0.058 to 0.313; model 4), but not significantly associated with the hypodivergent pattern after adjusting all considered covariates. The nasolabial angle was related to skeletal patterns in model 1, but no significant associations were found in the fully adjusted model. Likewise, the correlations of upper lip length with skeletal patterns were not significant in the final adjusted model, except for that with the hypodivergent pattern (β = 0.203, 95% CI 0.021 to 0.386; model 4). As for upper lip thickness and basic upper lip thickness, there was no significant association with any skeletal pattern after completely adjusted.

### 3.4. Stratified Associations between Upper Lip Characteristics and Skeletal Patterns

To obtain a further detailed distribution of upper lip characteristics among different skeletal subgroups, we combined sagittal and vertical skeletal patterns into nine subgroups, and compared the upper lip characteristics adjusted by linear regression (Figure 5 and Appendix A). Apparently shown in Figure 5, upper lip thickness, basic upper lip thickness, and superior sulcus depth were largest in Class III hypo-divergence, and smallest in Class II hyper-divergence. Both the upper lip length and nasolabial angle were smallest in Class III hypo-divergence, while the largest upper lip length and nasolabial angle were found in Class II hyperdivergent and Class II normodivergent participants, respectively.

Stratified association was further analyzed in the nine subgroups (Table 3). Superior sulcus depth was negatively correlated with the Class II hypodivergent pattern (β = −0.196, 95% CI −0.317 to −0.074; model 4), and positively related to the Class III normodivergent pattern (β = 0.471, 95% CI 0.128 to 0.813; model 4), the Class III hyperdivergent pattern (β = 0.355, 95% CI 0.172 to 0.537; model 4), and the Class III hypodivergent pattern (β = 0.198, 95% CI 0.041 to 0.354; model 4). The nasolabial angle was significantly negatively related to Class I hypodivergent (β = −1.348, 95% CI −2.670 to −0.026; model 4) and Class II hypodivergent pattern (β = −1.432, 95% CI −2.652 to −0.213; model 4). Upper lip length showed a significantly positive association only with the Class II hyperdivergent pattern (β = 0.434, 95% CI 0.106 to 0.763; model 4). Furthermore, basic upper lip thickness was only positively correlated with the Class III hyperdivergent pattern (β = 0.405, 95% CI 0.140 to 0.670; model 4). No significant relationship was found between upper lip thickness and any skeletal pattern subgroup after fully adjusted.

## 4. Discussion

The nose, lip, and chin are three key regions that determine profile esthetics on which orthodontic and orthognathic treatment focus [24]. In order to establish more accurate diagnoses and individualized post-treatment targets, it is essential to understand the associations of soft tissue characteristics with the morphology of adjacent dentoskeletal tissue. Current studies on nose and chin soft tissue morphology and its relationship with dentoskeletal patterns are abundant, while quite few studies were directly concerned with the lip [2,7,25,26]. Existing evidence has demonstrated that lip plays a crucial role in facial esthetics at both static and dynamic states, which attracts people’s attention at first sight during daily communication and interaction [12,17,27]. Thus, we aimed to pay specialized attention to the upper lip morphology and its correlation with skeletal patterns. The purpose of the study was to (1) establish the mean value of upper lip characteristics in Western China population; (2) compare upper lip characteristics in different sagittal and vertical skeletal pattern groups; and (3) explore the skeletal patterns which independently affects upper lip characteristics despite of all confounding variates.

Our study showed the prevalence of various skeletal patterns and mean values of upper lip characteristics including length, thickness, depth, and the nasolabial angle in Western China population. This is of great significance since there is a visible difference in hard and soft tissue characteristics among various races. Most of existing cephalometric analysis were originally developed by orthodontics in western countries, thus the reference values of cephalometric indices were standardized mainly according to Caucasians [28,29]. Nowadays, increased studies on the distribution of different skeletal patterns and standard values of cephalometric indices have been conducted in different countries and regions [2,26,30,31], hence it is necessary for us to develop the first study with adequate sample size in Chinese population.

Soft tissues in both nose and chin areas have been shown to significantly vary among different skeletal patterns [2,7,25,26]. Likewise, we also demonstrated that upper lip characteristics were significantly different in sagittal and vertical skeletal patterns, except for upper lip thickness among vertical skeletal patterns. Class II group was found to have a significantly larger nasolabial angle and upper lip length, and significantly smaller superior sulcus depth, basic upper lip thickness, and upper lip thickness. In addition, the five upper lip characteristics in Class III group changed inversely to that in Class II. It is conceivable that upper lip length increases due to maxillary overgrowth, and upper lip thickness augments in cases with maxillary retrusion due to soft tissue compensation, which is also confirmed by previous studies in other countries [5,32,33]. However, the significant decrease in nasolabial angle from Class II to Class III was contradictory to previous studies, including Burstone’s perspective that decreased nasolabial angle indicated maxillary protrusion [31,34,35]. In vertical dimension, the hyperdivergent group had a significantly larger nasolabial angle and upper lip length, and a significantly smaller superior sulcus depth and basic upper lip thickness, while the hypodivergent group exhibited opposite upper lip characteristics. This may be attributed to upper lip muscle tone and stiffness, which was reported less elastic and stiffer in hyperdivergent individuals compared to hypodivergent individuals [36]. Thus, a tense upper lip exhibited smaller superior sulcus depth and basic upper lip thickness in hyperdivergent participants. Notably, the upper lip characteristics were similar in Class II and hyperdivergent groups or Class III and hypodivergent groups, indicating the close relationship between corresponding sagittal and vertical skeletal patterns. A previous study showed that soft tissue thickness in the upper lip was significantly smaller in the high-angle group. However, the statistically significant association only exists in women [37]. Likewise, Perovic et al. have shown that upper lip thickness did not significantly differ dependently on vertical skeletal patterns, and that gender difference was established in this area [6]. Consistently in our study, upper lip thickness was not significantly different in three vertical skeletal pattern groups even after adjustment of all considered covariates including gender, indicating that the change in upper lip thickness was not dependent on vertical skeletal patterns.

Moreover, we evaluated the independent association of upper lip characteristics with different skeletal patterns, taking into account intermediate confounding factors. Multiple variates have been proven to influence soft tissue morphology. Gender is one of the most often-mentioned variates, with multiple studies showing that male patients tend to have thicker soft tissue compared with females with the same skeletal patterns, but the significance of gender difference varies among different skeletal patterns [6,10,38]. Race, as abovementioned, is an important genetic factor influencing soft tissue responding to skeletal patterns [30]. Age-related difference in soft tissue has also been commonly involved [2,39]. Particularly, the upper incisor position is another confounder emphasized in numerous studies, which exhibited significant associations of upper lip thickness and length with maxillary incisor protrusion [11,12]. Moreover, severe skeletal asymmetry or malformations may also affect upper lip characteristics, such as Pierre Robin syndrome which leads to a deformed upper lip [40].

After considering existing variates which have potential confounding effects on the relationship between upper lip characteristics and skeletal patterns, it was revealed that superior sulcus depth was negatively related to Class II, positively related to Class III and the hypodivergent pattern, and upper lip length was positively related to the hypodivergent pattern regardless of age, gender, molar relationship, crowding, overbite, overjet, upper incisor position, and other upper lip characteristics. The association of nasolabial angle, upper lip thickness, and length with skeletal patterns has been investigated by previous studies with controversial perspectives [2,19,26,31], which did not include adequate sample size nor adjust other confounding factors. This indicated that only abnormal superior sulcus depth might be directly and significantly corrected by early intervention of skeletal growth patterns. Specifically, inhibiting maxillary overgrowth in Class II might significantly decrease superior sulcus depth, while promoting maxillary growth in Class III might significantly increase superior sulcus depth. In terms of improving other upper lip characteristics, plenty of confounding factors should be taken into consideration instead of merely interfering skeletal growth.

To our knowledge, this study is the first evaluation of upper lip characteristics in Chinese population, which provides a significant reference for Chinese orthodontics in diagnosis and treatment planning. Furthermore, our study is the one evaluating relationship of upper lip morphology and skeletal patterns with a much larger sample size compared with similar previous studies [2,7,25,26], which confirmed the reliability of our results. A large sample size allows us to conduct multivariate linear regression to fully adjust the covariates, which is a common statistical method used in real-world studies [41,42,43], thus increasing the accuracy and authenticity of our comparison and association analysis. Moreover, the subgroup analysis combining sagittal and vertical patterns provided a more thorough and in-depth map of relationship between upper lip characteristics and skeletal patterns, which may guide early intervention in specific skeletal patterns for improvement in lip morphology. In addition, the associations of upper lip characteristics with the covariates adjusted in our study, such as age, overjet, and overbite, were also clinically instructive, which were shown in our supplementary materials. Notably, we only focused on the upper lip in this study due to the complexity in explanation of our comparison and association analysis results; further investigations into lower lip morphology and its relative position to the upper lip are needed.

## 5. Conclusions

The mean value of upper lip characteristics and prevalence of skeletal patterns were explored in Western China population, providing a race- and region-specific reference for Chinese orthodontists.Significant differences in upper lip characteristics were confirmed in various sagittal and vertical skeletal patterns, except for upper lip thickness among vertical skeletal patterns. This helps orthodontists make individualized treatment plans on the improvement of lip morphology according to patients with different skeletal patterns.The evaluation of independent correlation between upper lip characteristics and skeletal patterns provides guidance for a prognosis prediction of early intervention of skeletal growth patterns. Superior sulcus depth was negatively related to Class II, and positively related to Class III and the hypodivergent pattern regardless of age, gender, molar relationship, crowding, overbite, overjet, upper incisor position, and other upper lip characteristics, indicating that superior sulcus depth is the only upper lip index independently associated with skeletal patterns, and might be significantly corrected by intervention of skeletal growth. On the other hand, other upper lip characteristics may not be effectively improved by skeletal growth intervention, considering impacts of multiple covariates.

## Figures and Tables

**Figure 1 diagnostics-11-01713-f001:**
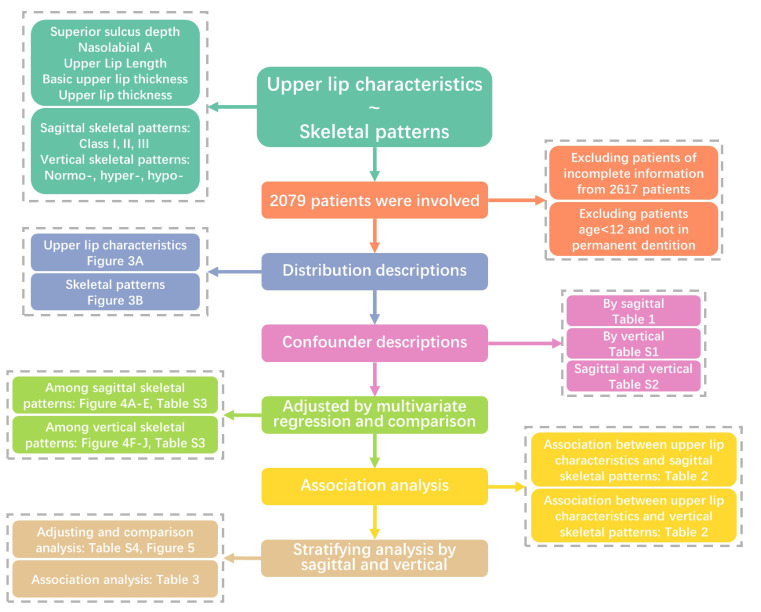
The flowchart demonstrates the investigation and analysis process.

**Figure 2 diagnostics-11-01713-f002:**
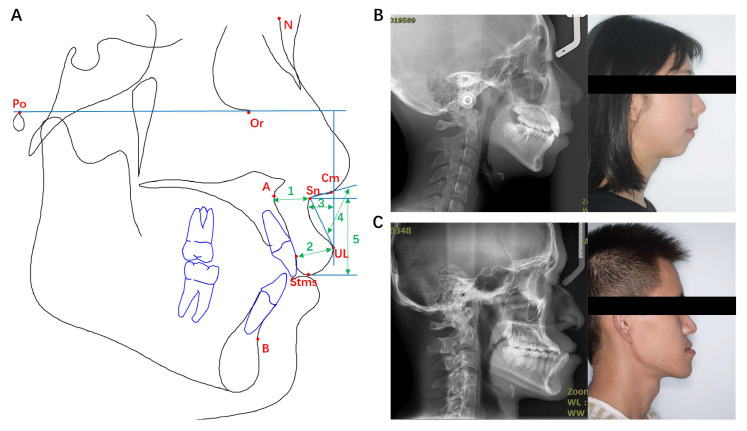
The 5 upper lip characteristics schematic diagrams used in the present study and representative patients’ profiles. (**A**) The 5 upper lip characteristics. 1: Basic upper lip thickness (mm); 2: Upper lip thickness (mm); 3: Superior sulcus depth (mm); 4: Nasolabial Angle (°); 5: Upper Lip Length (ULL) (mm). (**B**) A representative lateral X-ray and profile photo from a sagittal class II and vertical hyperdivergent patient. (**C**) A representative lateral X-ray and profile photo from a sagittal class III and vertical hypodivergent patient.

**Figure 3 diagnostics-11-01713-f003:**
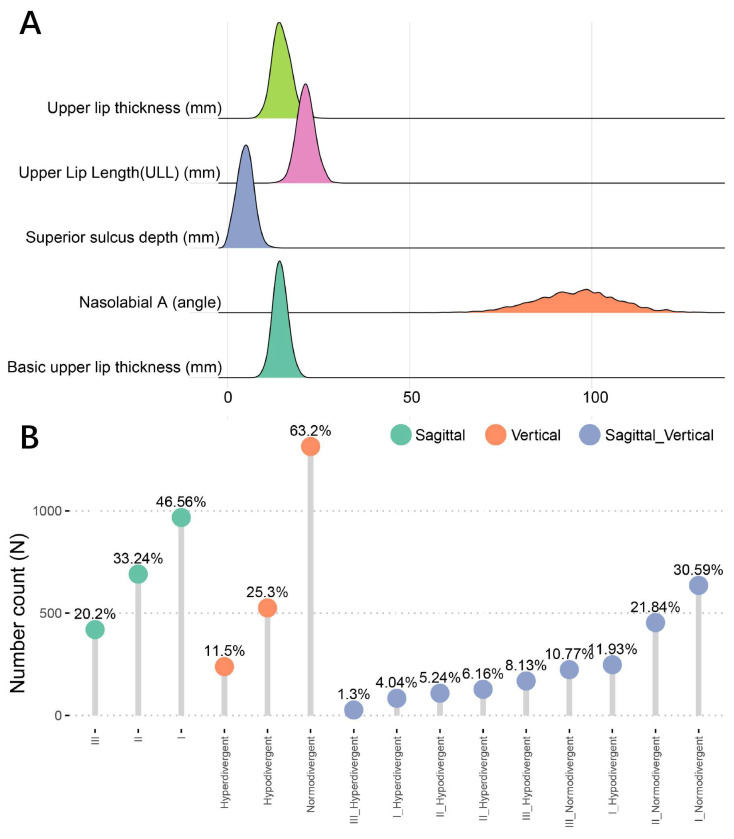
The data distribution of upper lip characteristics and skeletal patterns. (**A**) The distribution map shows the means and deviations of five upper lip characteristics. (**B**) The prevalence of different sagittal and vertical skeletal patterns.

**Figure 4 diagnostics-11-01713-f004:**
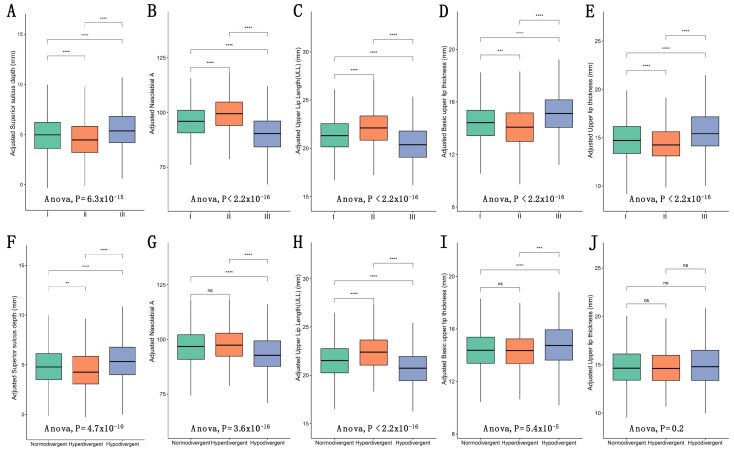
The comparison of adjusted upper lip characteristics among different sagittal (**A**–**E**) and vertical skeletal patterns (**F**–**J**). **: *p* < 0.01, ***: *p* < 0.001, ****: *p* < 0.0001, ns: not significant.

**Figure 5 diagnostics-11-01713-f005:**
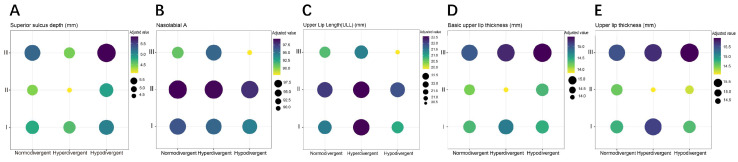
The comparisons of superior sulcus depth (**A**), nasolabial angle (**B**), upper lip length (**C**), basic upper lip thickness (**D**) and upper lip thickness (**E**) among different combinations of sagittal and vertical patterns.

**Table 1 diagnostics-11-01713-t001:** Demographic and clinical characteristics of samples involved in this study stratified by sagittal skeletal pattern.

Level	Overall	I	II	III	*p*-Value
N	2079	968	691	420	
Age (%)					
Adolescent	734 (35.3)	373 (38.5)	248 (35.9)	113 (26.9)	<0.001
Young adult	1272 (61.2)	566 (58.5)	409 (59.2)	297 (70.7)	
Middle age	73 (3.5)	29 (3.0)	34 (4.9)	10 (2.4)	
Gender (%)					
Male	623 (30.0)	271 (28.0)	199 (28.8)	153 (36.4)	0.005
Female	1456 (70.0)	697 (72.0)	492 (71.2)	267 (63.6)	
Molar Relationship (%)					
I	654 (31.5)	376 (38.8)	186 (26.9)	92 (21.9)	<0.001
II-1	654 (31.5)	288 (29.8)	318 (46.0)	48 (11.4)	
II-2	205 (9.9)	77 (8.0)	120 (17.4)	8 (1.9)	
III	506 (24.3)	197 (20.4)	52 (7.5)	257 (61.2)	
IV	60 (2.9)	30 (3.1)	15 (2.2)	15 (3.6)	
Upper crowding (%)					
I	1275 (61.3)	590 (61.0)	446 (64.5)	239 (56.9)	0.108
II	507 (24.4)	245 (25.3)	152 (22.0)	110 (26.2)	
III	297 (14.3)	133 (13.7)	93 (13.5)	71 (16.9)	
Lower crowding (%)					
I	1325 (63.7)	608 (62.8)	392 (56.7)	325 (77.4)	<0.001
II	557 (26.8)	270 (27.9)	210 (30.4)	77 (18.3)	
III	197 (9.5)	90 (9.3)	89 (12.9)	18 (4.3)	
Overbite (%)					
Normal	1302 (62.6)	647 (66.8)	463 (67.0)	192 (45.7)	<0.001
Deep	246 (11.8)	91 (9.4)	112 (16.2)	43 (10.2)	
Open	117 (5.6)	40 (4.1)	39 (5.6)	38 (9.0)	
Shallow	414 (19.9)	190 (19.6)	77 (11.1)	147 (35.0)	
Overjet (%)					
Normal	1125 (54.1)	638 (65.9)	299 (43.3)	188 (44.8)	<0.001
Cross	234 (11.3)	36 (3.7)	5 (0.7)	193 (46.0)	
Deep	690 (33.2)	284 (29.3)	385 (55.7)	21 (5.0)	
Shallow	30 (1.4)	10 (1.0)	2 (0.3)	18 (4.3)	
Vertical skeletal pattern (%)					
Normodivergent	1314 (63.2)	636 (65.7)	454 (65.7)	224 (53.3)	<0.001
Hyperdivergent	239 (11.5)	84 (8.7)	128 (18.5)	27 (6.4)	
Hypodivergent	526 (25.3)	248 (25.6)	109 (15.8)	169 (40.2)	
U1-ANS (mm) (mean (SD))	27.60 (3.08)	27.54 (2.93)	28.50 (2.91)	26.29 (3.21)	<0.001
U1-OP (mean (SD))	54.09 (7.65)	54.18 (7.42)	54.33 (8.28)	53.47 (7.06)	0.168
U1-PP (mm) (mean (SD))	27.28 (3.12)	27.22 (2.97)	28.18 (2.96)	25.94 (3.22)	<0.001
U1-NA (mean (SD))	27.92 (8.76)	28.06 (8.12)	24.92 (9.08)	32.55 (7.49)	<0.001
U1-NA (mm) (mean (SD))	5.65 (2.85)	5.72 (2.76)	4.78 (2.61)	6.92 (2.93)	<0.001
U1-SN (mean (SD))	108.46 (9.48)	108.40 (8.99)	106.34 (10.12)	112.08 (8.37)	<0.001
U1-PP (mean (SD))	119.54 (8.88)	119.36 (8.54)	117.94 (9.43)	122.57 (7.93)	<0.001
U1-NPo (mm) (mean (SD))	9.86 (4.92)	9.42 (3.71)	13.60 (4.16)	4.74 (3.17)	<0.001
UL-EL (mm) (mean (SD))	0.53 (2.85)	0.45 (2.32)	2.35 (2.32)	−2.28 (2.39)	<0.001
UL-SL (mean (SD))	4.60 (2.64)	4.65 (2.57)	4.24 (2.52)	5.07 (2.90)	<0.001
Upper Lip Length (ULL) (mm) (mean (SD))	21.46 (2.40)	21.41 (2.24)	22.16 (2.30)	20.41 (2.52)	<0.001
Basic upper lip thickness (mm) (mean (SD))	14.48 (1.97)	14.42 (1.82)	14.14 (2.00)	15.18 (2.07)	<0.001
Nasolabial A (mean (SD))	95.74 (11.83)	95.80 (11.34)	99.17 (11.02)	89.96 (12.01)	<0.001
Upper lip thickness (mm) (mean (SD))	14.85 (2.50)	14.80 (2.43)	14.40 (2.36)	15.69 (2.67)	<0.001
Superior sulcus depth (mm) (mean (SD))	4.84 (2.17)	4.85 (2.16)	4.44 (2.14)	5.45 (2.11)	<0.001

**Table 2 diagnostics-11-01713-t002:** Association of skeletal patterns with upper lip characteristics by multivariate analysis.

	β (95% CI)			
	Model 1 ^a^	Model 2 ^b^	Model 3 ^c^	Model 4 ^d^
Superior sulcus depth (mm)				
SSP-Class I	1 [Reference]	1 [Reference]	1 [Reference]	1 [Reference]
SSP-Class II	−0.300 ** (−0.516, −0.084)	−0.587 *** (−0.827, −0.346)	−0.471 *** (−0.569, −0.372)	−0.195 *** (−0.302, −0.087)
SSP-Class III	0.301 * (0.011, 0.590)	0.613 *** (0.331, 0.896)	0.752 *** (0.630, 0.875)	0.253 *** (0.117, 0.389)
VSP-Normodivergent	1 [Reference]	1 [Reference]	1 [Reference]	1 [Reference]
VSP-Hyperdivergent	−0.134 (−0.427, 0.159)	0.150 (−0.141, 0.441)	−0.114 (−0.241, 0.014)	0.185 ** (0.058, 0.313)
VSP-Hypodivergent	0.307 ** (0.086, 0.527)	0.143 (−0.088, 0.375)	0.267 *** (0.171, 0.362)	0.050 (−0.052, 0.151)
Nasolabial A				
SSP-Class I	1 [Reference]	1 [Reference]	1 [Reference]	1 [Reference]
SSP-Class II	2.729 *** (1.544, 3.914)	2.255 ** (0.839, 3.670)	−1.446 ** (−2.378, −0.515)	−0.922 (−1.998, 0.154)
SSP-Class III	−3.754 *** (−5.341, −2.168)	−3.907 *** (−5.565, −2.248)	1.219 * (0.050, 2.387)	0.524 (−0.839, 1.887)
VSP-Normodivergent	1 [Reference]	1 [Reference]	1 [Reference]	1 [Reference]
VSP-Hyperdivergent	−0.103 (−1.712, 1.506)	−1.346 (−3.056, 0.365)	−1.389 * (−2.568, −0.210)	−0.346 (−1.622, 0.930)
VSP-Hypodivergent	−1.888 ** (−3.097, −0.678)	−1.014 (−2.375, 0.347)	0.165 (−0.723, 1.053)	−0.582 (−1.599, 0.434)
Upper Lip Length (ULL) (mm)				
SSP-Class I	1 [Reference]	1 [Reference]	1 [Reference]	1 [Reference]
SSP-Class II	0.536 *** (0.318, 0.754)	−0.011 (−0.233, 0.210)	0.427 *** (0.197, 0.657)	−0.043 (−0.236, 0.150)
SSP-Class III	−0.901 *** (−1.193, −0.609)	−0.148 (−0.407, 0.111)	−0.732 *** (−1.020, −0.445)	−0.176 (−0.421, 0.068)
VSP-Normodivergent	1 [Reference]	1 [Reference]	1 [Reference]	1 [Reference]
VSP-Hyperdivergent	0.772 *** (0.476, 1.068)	−0.019 (−0.286, 0.248)	0.723 *** (0.432, 1.013)	−0.069 (−0.298, 0.160)
VSP-Hypodivergent	−0.689 *** (−0.911, −0.466)	0.315 ** (0.102, 0.527)	−0.629 *** (−0.846, −0.411)	0.203 * (0.021, 0.386)
Basic upper lip thickness (mm)				
SSP-Class I	1 [Reference]	1 [Reference]	1 [Reference]	1 [Reference]
SSP-Class II	−0.457 *** (−0.639, −0.275)	0.060 (−0.174, 0.294)	−0.741 *** (−0.912, −0.570)	−0.141 (−0.297, 0.015)
SSP-Class III	0.721 *** (0.477, 0.964)	0.299 * (0.025, 0.573)	1.333 *** (1.123, 1.543)	0.194 (−0.004, 0.392)
VSP-Normodivergent	1 [Reference]	1 [Reference]	1 [Reference]	1 [Reference]
VSP-Hyperdivergent	0.278 * (0.032, 0.525)	0.0001 (−0.282, 0.283)	−0.376 *** (−0.595, −0.157)	0.179 (−0.005, 0.364)
VSP-Hypodivergent	−0.069 (−0.254, 0.117)	0.278 * (0.053, 0.503)	0.588 *** (0.425, 0.751)	0.111 (−0.036, 0.258)
Upper lip thickness (mm)				
SSP-Class I	1 [Reference]	1 [Reference]	1 [Reference]	1 [Reference]
SSP-Class II	−0.250 * (−0.485, −0.015)	−0.480 ** (−0.769, −0.191)	0.159 (−0.074, 0.392)	−0.109 (−0.298, 0.079)
SSP-Class III	0.193 (−0.122, 0.508)	0.959 *** (0.620, 1.298)	−0.178 (−0.470, 0.114)	0.194 (−0.045, 0.433)
VSP-Normodivergent	1 [Reference]	1 [Reference]	1 [Reference]	1 [Reference]
VSP-Hyperdivergent	0.422 ** (0.102, 0.741)	−0.052 (−0.401, 0.298)	0.386 * (0.092, 0.680)	−0.089 (−0.313, 0.135)
VSP-Hypodivergent	−0.332 ** (−0.572, −0.092)	0.115 (−0.164, 0.393)	−0.455 *** (−0.676, −0.234)	−0.077 (−0.256, 0.101)

Note: * *p* < 0.05; ** *p* < 0.01; *** *p* < 0.001. Abbreviation: SSP, Sagittal skeletal pattern; VSP, Vertical skeletal pattern; CI, confidence interval. ^a^ Model 1 was adjusted for “Age”, “Gender”, “Molar Relationship”, “Upper crowding”, “Lower crowding”, “Overbite” and “Overjet”. ^b^ Model 2 was adjusted for “U1-ANS (mm)”, “U1-OP”, “U1-PP (mm)”, “U1-NA”, “U1-NA (mm)”, “U1-SN”, “U1-PP” and “U1-NPo (mm)”. ^c^ Model 3 was adjusted for “Nasolabial A”, “UL-EP (mm)”, “Upper lip to S line”, “Upper Lip Length (ULL) (mm)”, “Basic upper lip thickness (mm)” and “Upper lip thickness (mm)”. ^d^ Model 4 was adjusted for all of above variables.

**Table 3 diagnostics-11-01713-t003:** Association of skeletal patterns with upper lip characteristics by multivariate analysis (stratified by sagittal and vertical skeletal combination patterns).

	β (95% CI)			
	Model 1 ^a^	Model 2 ^b^	Model 3 ^c^	Model 4 ^d^
Superior sulcus depth (mm)				
I-Norm	1 [Reference]	1 [Reference]	1 [Reference]	1 [Reference]
I-Hyper	−0.037 (−0.506, 0.433)	0.341 (−0.109, 0.791)	−0.089 (−0.297, 0.119)	0.190 (−0.004, 0.384)
I-Hypo	0.241 (−0.067, 0.548)	0.024 (−0.280, 0.327)	0.232 *** (0.098, 0.366)	0.007 (−0.125, 0.139)
II-Norm	−0.378 (−0.782, 0.027)	−0.446 * (−0.878, −0.014)	−0.686 *** (−0.868, −0.504)	−0.035 (−0.224, 0.153)
II-Hyper	−0.047 (−0.480, 0.385)	−0.487 * (−0.902, −0.073)	−0.149 (−0.334, 0.037)	−0.173 (−0.357, 0.010)
II-Hypo	−0.317 * (−0.575, −0.058)	−0.570 *** (−0.845, −0.295)	−0.469 *** (−0.587, −0.352)	−0.196 ** (−0.317, −0.074)
III-Norm	−0.515 (−1.344, 0.314)	0.185 (−0.582, 0.953)	0.978 *** (0.623, 1.333)	0.471 ** (0.128, 0.813)
III-Hyper	0.709 *** (0.317, 1.101)	0.907 *** (0.506, 1.308)	1.016 *** (0.842, 1.191)	0.355 *** (0.172, 0.537)
III-Hypo	0.277 (−0.070, 0.623)	0.510 ** (0.167, 0.853)	0.708 *** (0.558, 0.858)	0.198 * (0.041, 0.354)
Nasolabial A				
I-Norm	1 [Reference]	1 [Reference]	1 [Reference]	1 [Reference]
I-Hyper	−0.940 (−3.510, 1.631)	−2.155 (−4.797, 0.488)	−1.699 (−3.622, 0.224)	−1.041 (−2.981, 0.900)
I-Hypo	−1.830 * (−3.514, −0.145)	−0.758 (−2.540, 1.025)	−0.241 (−1.481, 1.000)	−1.348 * (−2.670, −0.026)
II-Norm	2.006 (−0.210, 4.222)	0.088 (−2.449, 2.626)	−3.719 *** (−5.414, −2.024)	−1.751 (−3.635, 0.133)
II-Hyper	1.528 (−0.843, 3.899)	2.085 (−0.349, 4.519)	−0.267 (−1.984, 1.450)	−0.891 (−2.725, 0.944)
II-Hypo	2.645 *** (1.228, 4.063)	1.995 * (0.378, 3.611)	−1.696 ** (−2.797, −0.594)	−1.432 * (−2.652, −0.213)
III-Norm	1.417 (−3.127, 5.962)	0.052 (−4.456, 4.559)	3.714 * (0.412, 7.016)	2.049 (−1.377, 5.475)
III-Hyper	−6.291 *** (−8.439, −4.143)	−5.712 *** (−8.068, −3.356)	0.970 (−0.694, 2.635)	−0.013 (−1.846, 1.820)
III-Hypo	−3.855 *** (−5.755, −1.956)	−3.565 *** (−5.577, −1.552)	0.923 (−0.490, 2.337)	0.029 (−1.539, 1.597)
Upper Lip Length (ULL) (mm)				
I-Norm	1 [Reference]	1 [Reference]	1 [Reference]	1 [Reference]
I-Hyper	1.031 *** (0.558, 1.505)	0.108 (−0.305, 0.521)	0.997 *** (0.523, 1.471)	−0.032 (−0.380, 0.316)
I-Hypo	−0.663 *** (−0.973, −0.353)	0.210 (−0.069, 0.488)	−0.575 *** (−0.881, −0.270)	0.104 (−0.133, 0.341)
II-Norm	1.097 *** (0.689, 1.505)	−0.196 (−0.592, 0.200)	0.966 *** (0.547, 1.385)	−0.237 (−0.575, 0.102)
II-Hyper	0.102 (−0.335, 0.539)	0.664 *** (0.284, 1.044)	0.154 (−0.270, 0.579)	0.434 ** (0.106, 0.763)
II-Hypo	0.574 *** (0.313, 0.835)	−0.117 (−0.369, 0.136)	0.443 ** (0.170, 0.715)	−0.141 (−0.360, 0.078)
III-Norm	0.306 (−0.531, 1.143)	−0.062 (−0.766, 0.641)	0.311 (−0.506, 1.128)	−0.103 (−0.718, 0.512)
III-Hyper	−1.688 *** (−2.084, −1.293)	0.060 (−0.308, 0.427)	−1.544 *** (−1.950, −1.137)	−0.055 (−0.384, 0.273)
III-Hypo	−0.805 *** (−1.155, −0.456)	−0.051 (−0.365, 0.264)	−0.549 ** (−0.898, −0.200)	−0.128 (−0.409, 0.153)
Basic upper lip thickness (mm)				
I-Norm	1 [Reference]	1 [Reference]	1 [Reference]	1 [Reference]
I-Hyper	0.464 * (0.070, 0.858)	0.278 (−0.159, 0.715)	−0.297 (−0.655, 0.062)	0.234 (−0.048, 0.515)
I-Hypo	−0.155 (−0.414, 0.103)	0.175 (−0.119, 0.470)	0.420 *** (0.189, 0.650)	0.071 (−0.121, 0.263)
II-Norm	−0.378 * (−0.718, −0.038)	−0.147 (−0.567, 0.272)	−1.242 *** (−1.555, −0.929)	0.025 (−0.249, 0.299)
II-Hyper	−0.551 ** (−0.914, −0.187)	0.338 (−0.064, 0.741)	−0.107 (−0.427, 0.213)	−0.121 (−0.387, 0.145)
II-Hypo	−0.411 *** (−0.629, −0.194)	0.108 (−0.159, 0.375)	−0.778 *** (−0.981, −0.575)	−0.114 (−0.291, 0.063)
III-Norm	1.269 *** (0.573, 1.966)	0.384 (−0.361, 1.129)	0.962 ** (0.347, 1.576)	0.273 (−0.224, 0.770)
III-Hyper	0.804 *** (0.475, 1.133)	0.728 *** (0.339, 1.118)	2.032 *** (1.735, 2.330)	0.405 ** (0.140, 0.670)
III-Hypo	0.599 *** (0.308, 0.891)	0.184 (−0.148, 0.517)	1.194 *** (0.935, 1.452)	0.117 (−0.110, 0.345)
Upper lip thickness (mm)				
I-Norm	1 [Reference]	1 [Reference]	1 [Reference]	1 [Reference]
I-Hyper	0.899 *** (0.389, 1.410)	0.464 (−0.075, 1.003)	0.869 *** (0.390, 1.349)	−0.018 (−0.359, 0.324)
I-Hypo	−0.247 (−0.582, 0.087)	−0.059 (−0.422, 0.305)	−0.396 * (−0.705, −0.087)	−0.119 (−0.352, 0.113)
II-Norm	−0.001 (−0.441, 0.439)	−0.748 ** (−1.265, −0.230)	0.284 (−0.141, 0.709)	−0.251 (−0.582, 0.080)
II-Hyper	−0.747 ** (−1.217, −0.276)	−0.506 * (−1.003, −0.009)	−0.393 (−0.822, 0.036)	−0.185 (−0.507, 0.137)
II-Hypo	−0.070 (−0.351, 0.211)	−0.361 * (−0.691, −0.031)	0.336 * (0.060, 0.611)	−0.105 (−0.320, 0.109)
III-Norm	0.436 (−0.466, 1.338)	0.440 (−0.480, 1.359)	0.428 (−0.398, 1.254)	0.090 (−0.513, 0.692)
III-Hyper	0.061 (−0.365, 0.488)	1.413 *** (0.932, 1.894)	−0.468 * (−0.884, −0.053)	0.168 (−0.154, 0.490)
III-Hypo	0.197 (−0.180, 0.574)	0.731 *** (0.321, 1.142)	−0.205 (−0.559, 0.148)	0.159 (−0.116, 0.435)

Note: * *p* < 0.05; ** *p* < 0.01; *** *p* < 0.001. Abbreviation: CI, confidence interval. ^a^ Model 1 was adjusted for “Age”, “Gender”, “Molar Relationship”, “Upper crowding”, “Lower crowding”, “Overbite” and “Overjet”. ^b^ Model 2 was adjusted for “U1-ANS (mm)”, “U1-OP”, “U1-PP (mm)”, “U1-NA”, “U1-NA (mm)”, “U1-SN”, “U1-PP” and “U1-NPo (mm)”. ^c^ Model 3 was adjusted for “Nasolabial A”, “UL-EP (mm)”, “Upper lip to S line”, “UL-Sn vert (mm)”, “Upper Lip Length (ULL) (mm)”, “Basic upper lip thickness (mm)” and “Upper lip thickness (mm)”. ^d^ Model 4 was adjusted for all of above variables.

## Data Availability

Not applicable.

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
