# Peer review of "Association of Upper Lip Morphology Characteristics with Sagittal and Vertical Skeletal Patterns: A Cross Sectional Study"

_diagnostics, 2021, doi:10.3390/diagnostics11091713_

Round 1

Reviewer 1 Report

Regarding the present article, I have a positive impression about the huge amount of data and the complex presentation. 

I think that some elements which increase the interest among readers could be included.

Author Response

Thank you for the professional suggestion and we feel sorry not to well demonstrate clinical significance of our results, which is the most interesting part for readers. Therefore, we have added clinical relevance of our results in Discussion paragraphs, and we made a focused summary of clinical interest of this study in Conclusion part considering discrete distribution of our previous discussion (Line 407-422). We hope this will improve the clarity of clinical meanings of our study to readers so they could use the conclusion directly.

Reviewer 2 Report

I revised the paper "Association of upper lip morphology characteristics with sagittal and vertical skeletal patterns: a cross sectional study". Some points should be clarified:

Introduction

  • The purpose paragraph should contain a null hypothesis sentence before the specific purpose sentence of the study.

methods

  • "Written informed consent was obtained from every participant". Considering the age of the included study population, I think not all participants can sign informed consent.

  • Line 127 needs a reference: "Class I is defined as ANB ≥ 1 ° and ≤ 5 °, class II is defined as ANB> 5 ° and class III is defined as ANB <1 °"

  • Line 147: “IV when the molar ratio is II on one side and III on the other.” Have you evaluated the skeletal asymmetry problems? There are no details in the inclusion or exclusion criteria. Also, how did you consider the subdivision of Class II?

  • Lines 148-150 need a reference: “Crowding is rated as I (<4mm), II (4-8mm) and III (≥ 8mm). The overbite is divided into open (<0 mm), superficial (0-1 mm), normal (1-5 mm) and deep (> 5 mm). Overjet is divided into cross (<0mm), superficial (0-1mm), normal (1-5mm) and deep (> 5mm)

  • Did you perform a sample size calculation?
  • Have you calculated the intra-examiner reliability of the assessments?
  • Line 182: "Among the participants included in this study, the five cephalometric indices of the characteristics of the upper lip were normally distributed." Did you perform the Kolmogorov-Smirnov test? Did you base your sentence on the central limit theorem? It should be specified.

  • 3 A and B should have a more detailed legend.

Results

  • In the Results section, you should avoid repeating the data shown in the tables.
  • 4: You should better describe each box plot. Considering that you show bivariate comparisons, you should emphasize the mean along with the quartiles.
  • In multivariate analysis, the values ​​of the intercept and the value of R squared must be added.

Discussion

  • Line 291: " Nose, lip and chin are three key regions determining profile esthetics on which orthodontic and orthognathic treatment focus on". I suggest citing a recent review on this topic: Barone, S., Morice, A., Picard, A., & Giudice, A. (2020). Orthognathic approach before surgery vs conventional orthognathic approach: a systematic review of systematic reviews. Journal of stomatology, oral and maxillofacial surgery.

  • The first paragraph of the Discussion section should emphasize the purpose of the study.

  • In the Discussion section you should also add comments about patients with skeletal asymmetry or skeletal malformations that may affect your results. I suggest citing Giudice, A., Barone, S., Belhous, K., Morice, A., Soupre, V., Bennardo, F., ... & Picard, A. (2018). Sequence by Pierre Robin: A Complete Narrative Review of Literature Over Time. Journal of Stomatology, Oral and Maxillofacial Surgery, 119 (5), 419-428.

  • Line 318: “[..] and upper lip thickness augments in cases with maxillary retrusion due to soft tissue compensation, which is also confirmed by previous studies in other countries [5, 29, 30]. I think the U1 proclination also plays an important role as a compensatory mechanism. You should discuss it.

  • Lines 323-325: " In vertical dimension, hyperdivergent group had significantly larger nasolabial angle and upper lip length, and significantly smaller superior sulcus depth and basic upper lip thickness, while hypodivergent group exhibited opposite upper lip characteristics.". You should comment on your data, not just report it. For these differences, I think that muscles play an important role that should be emphasized.

  • Lines 345-349: " After considering existing variates which have potential confounding effects on the relationship between upper lip characteristics and skeletal patterns, it was revealed that superior sulcus depth was negatively related to Class II, positively related to Class III and hypodivergent pattern, and upper lip length was positively related to hypodivergent pattern regardless of age, gender, molar relationship, crowding, overbite, overjet, upper incisor position and other upper lip characteristics.” Explain better the clinical relevance after reporting your results.

Author Response

Reviewer 2

Introduction

  1. The purpose paragraph should contain a null hypothesis sentence before the specific purpose sentence of the study.

Reply: Thanks for this suggestion. We have added a null hypothesis in the purpose paragraph of the revision (Line 80-81).

methods

  1. "Written informed consent was obtained from every participant". Considering the age of the included study population, I think not all participants can sign informed consent.

Reply: Thank you for reminding and we feel sorry to not well address this issue. We actually obtained informed consent from adult participant and the guardians of minors, and we revised the expression in the manuscript (Line 97).

  1. Line 127 needs a reference: "Class I is defined as ANB ≥ 1 ° and ≤ 5 °, class II is defined as ANB> 5 ° and class III is defined as ANB <1 °"

Reply: Thank you for your suggestion. The criterion for sagittal skeletal classification we used is mainly referred by two Chinese orthodontic textbooks:

[1] Fu Minkui, Lin Jiuxiang. Orthodontics [M]. Peking University Medical Press, 2014. (傅民魁, 林久祥. 口腔正畸学[M]. 北京大学医学出版社, 2014.)

[2] Chen Yangxi (Ed.). Orthodontics: Basic, Technical and clinical aspects [M]. People's Medical Publishing House, 2012. (陈扬熙(主编). 《口腔正畸学》——基础、技术与临床[M]. 人民卫生出版社, 2012.)

These two books are respectively used in the orthodontic teaching of undergraduate and graduate students in China, and are widely read by Chinese orthodontists, which has a very authoritative influence. The reference values of sagittal skeletal classification in these two books are based on the results of Chinese cephalometric measurement research, which is a more suitable standard for Chinese characteristics, though different from the ABO references. Considering one of our main purposes in the study is to describe the upper lip characteristics for Chinese, it will be more suitable to adapt a classification reference value for Chinese. We have referred the two books and added a description for this issue in the manuscript (Line 129-130).

  1. Line 147: “IV when the molar ratio is II on one side and III on the other.” Have you evaluated the skeletal asymmetry problems? There are no details in the inclusion or exclusion criteria. Also, how did you consider the subdivision of Class II?

Reply: Thanks for the professional advice.

We are sorry for not describing this detail in the manuscript. In fact, we specifically reviewed the data of 60 class IV patients and found that only a few patients had mild asymmetry in the length of the mandibular ramus, and no other obvious asymmetry was found. The main reason they were grouped to Class IV is due to the dental factor. The reason why we did not evaluate the effect of Class IV could be explained in the following perspectives. Firstly, the number of Class IV patients was very small (only accounting for 2.9%), which is less than the widely used statistical significance (5%), hence the effect of Class IV on the overall outcome was very weak. Secondly, the multivariate linear regression we used have adjusted the bias caused by Class IV, so the statistical results are reliable. Finally, the most important thing is that the five upper lip characteristics we paid attention to are the profile descriptions of the relationship mainly in sagittal and vertical direction, while the asymmetry is mainly the relationship in transverse direction, which has almost no influence on assessment of our five indicators. Moreover, facial asymmetry mainly occurs in the mandible, which may impact on the soft tissue appearance of the mandible while hardly on that of the maxilla.

As for Class II, we distinguished Class II-2 from Class II-1 based on the incisors retroclination (U1-SN < 100°), because retroclined incisors play an important role in upper lip characteristics.

  1. Lines 148-150 need a reference: “Crowding is rated as I (<4mm), II (4-8mm) and III (≥ 8mm). The overbite is divided into open (<0 mm), superficial (0-1 mm), normal (1-5 mm) and deep (> 5 mm). Overjet is divided into cross (<0mm), superficial (0-1mm), normal (1-5mm) and deep (> 5mm).

Reply: Thank you for your suggestion. This is a situation similar to the question 3. The criterion for crowding, overbite and overjet we used are mainly referred by two Chinese orthodontic textbooks:

[1] Fu Minkui, Lin Jiuxiang. Orthodontics [M]. Peking University Medical Press, 2014. (傅民魁, 林久祥. 口腔正畸学[M]. 北京大学医学出版社, 2014.)

[2] Chen Yangxi (Ed.). Orthodontics: Basic, Technical and clinical aspects [M]. People's Medical Publishing House, 2012. (陈扬熙(主编). 《口腔正畸学》——基础、技术与临床[M]. 人民卫生出版社, 2012.)

These two books are respectively used in the orthodontic teaching of undergraduate and graduate students in China, and are widely read by Chinese orthodontists, which has a very authoritative influence. The reference values for crowding, overbite and overjet in these two books are based on the results of Chinese plaster model research, which is a little different from the references from other countries. Considering one of our main purposes in the study is to describe the upper lip characteristics for Chinese, it will be more suitable to adapt a Chinese reference value. We have referred the two books and added a description for this issue in the manuscript (Line 153-154).

  1. Did you perform a sample size calculation?

Reply: Thank you for the advice. We conducted a population-based real-world analysis, we collected patients of a certain type in a certain place over a certain period of time from hospital electronic medical record system. Sample size estimates were not carried out. Since the time, place and disease were determined, the sample size was also determined. The retrospective nature of the study predetermines the sample size (Gauss et al., 2019; Pincus et al., 2017; Shindo et al., 2015). Besides, according to the statistic theory, if positive findings are found (P < 0.05), there is usually no need to worry about the sample size. If the result is negative, the sample size can be estimated. For example, if you see a flower bloom and take a photo of it, no one will question you. If you do not see the flowers bloom, you say the flowers will not bloom, people will question you because it may be an illusion caused by too few observations (small sample size). Furthermore, the sample size in our study is much larger than previous similar studies (thousands in this study vs. hundreds or dozens in previous studies).  

  1. Have you calculated the intra-examiner reliability of the assessments?

Reply: Thank you for your professional advice, we are sorry for our negligence in describing the intra-examiner reliability of the assessments. All the variables in the manuscript were detected by two examiners. For demographic characteristics (age and gender), they were directly exported from the hospital record system. For categorical variables (molar relationships), all authors were involved in discussion of the suitable classification if the initial results of two examiners were inconsistent, and a final consistent classification was obtained. For continuous variables, intraclass correlation coefficient (ICC) was used, and all the ICC values were > 0.9 with corresponding P-values < 0.05, indicating the consistency of intra-examiner agreement was very reliable, and then the average were calculated (Line 179-185). We have added these descriptions in the manuscript.

  1. Line 182: "Among the participants included in this study, the five cephalometric indices of the characteristics of the upper lip were normally distributed." Did you perform the Kolmogorov-Smirnov test? Did you base your sentence on the central limit theorem? It should be specified.

Reply: Thank you for your professional suggestion, and we are very sorry for our carelessness. We made a preliminary judgment that the distribution was normal only based on the figures, but we forgot to do statistical test. The writer and the analyst did not communicate this well. We apologize again and thank the reviewer for pointing this problem in time. We have made the Kolmogorov-Smirnov test as suggested, and found that only the P-value of upper lip thickness is <0.05, indicating upper lip thickness is not normally distributed, while the other four characteristics with P-values > 0.05 indicating they are normally distributed. We have amended this in the manuscript (Line 193-194).

  1. 3 A and B should have a more detailed legend.

Reply: Thanks for the suggestion. We specified the figure legends for Figure 3A and 3B, with the detailed data was described in the manuscript (Line 206-207).

Results

  1. In the Results section, you should avoid repeating the data shown in the tables.

Reply: Thanks for the advice. We deleted simple repetition of data in the manuscript as suggested (Line 194-196, Line 214-217), but remained the data in the brackets in order to support the description of results.

  1. You should better describe each box plot. Considering that you show bivariate comparisons, you should emphasize the mean along with the quartiles.

Reply: Thank you for providing a valuable perspective. Indeed, readers will be interested in specific values among groups. Due to there are a lot of results to report, we made these as readable tables. All the comparisons and their means etc. were presented in Table S3 as supplementary materials.

  1. In multivariate analysis, the values ​​of the intercept and the value of R squared must be added.

Reply: Thank you for kindly reminding. As for the multivariate analysis, the β (95% CI) and P-value are the most important results should be presented, as we demonstrated in the main manuscript. We agree with you that the intercept and the value of R squared are also important. Due to the limited layout of manuscript and abundant results, we provided the complete results of multivariate analysis including intercept and the value of R squared as supplementary materials, in which fully detailed information of every model can be found.

Discussion

  1. Line 291: " Nose, lip and chin are three key regions determining profile esthetics on which orthodontic and orthognathic treatment focus on". I suggest citing a recent review on this topic: Barone, S., Morice, A., Picard, A., & Giudice, A. (2020). Orthognathic approach before surgery vs conventional orthognathic approach: a systematic review of systematic reviews. Journal of stomatology, oral and maxillofacial surgery.

Reply: Thank you for this suggestion. We have added this reference in the revision (Line 307).

  1. The first paragraph of the Discussion section should emphasize the purpose of the study.

Reply: Thanks for the advice. We specified our purposes in the Discussion as suggested (Line 315-319).

  1. In the Discussion section you should also add comments about patients with skeletal asymmetry or skeletal malformations that may affect your results. I suggest citing Giudice, , Barone, S., Belhous, K., Morice, A., Soupre, V., Bennardo, F., ... & Picard, A. (2018). Sequence by Pierre Robin: A Complete Narrative Review of Literature Over Time. Journal of Stomatology, Oral and Maxillofacial Surgery, 119 (5), 419-428.

Reply: Thank the reviewer very much for the valuable suggestion. We agree with the reviewer’s point that patients with severe skeletal asymmetry or malformations may affect upper lip characteristics. We have discussed the issue in the manuscript and correctly cited the paper (Line 369-370).

  1. Line 318: “[..] and upper lip thickness augments in cases with maxillary retrusion due to soft tissue compensation, which is also confirmed by previous studies in other countries [5, 29, 30]. I think the U1 proclination also plays an important role as a compensatory mechanism. You should discuss it.

Reply: Thank you for providing a valuable perspective. Indeed, U1 proclination plays an important role in upper lip morphology, which was also discussed in many previous studies. However, U1 proclination was one of the confounding factors that were adjusted, and our study purely evaluated the association of upper lip morphology with skeletal patterns independently of other covariates, including U1 proclination. Thus, we explained the results only based on upper lip and skeletal pattern themselves. While the relationship between U1 position and upper lip morphology needs to be further investigated using another multivariate regression model, which we are subsequently working on.

  1. Lines 323-325: " In vertical dimension, hyperdivergent group had significantly larger nasolabial angle and upper lip length, and significantly smaller superior sulcus depth and basic upper lip thickness, while hypodivergent group exhibited opposite upper lip characteristics.". You should comment on your data, not just report it. For these differences, I think that muscles play an important role that should be emphasized.

Reply: Thank you very much for the instructive suggestion. We explained the results based on differences in upper lip muscle among skeletal patterns in the revision (Line 345-351).

  1. Lines 345-349: " After considering existing variates which have potential confounding effects on the relationship between upper lip characteristics and skeletal patterns, it was revealed that superior sulcus depth was negatively related to Class II, positively related to Class III and hypodivergent pattern, and upper lip length was positively related to hypodivergent pattern regardless of age, gender, molar relationship, crowding, overbite, overjet, upper incisor position and other upper lip characteristics.” Explain better the clinical relevance after reporting your results.

Reply: Thanks for the suggestion. As we discussed in the previous manuscript, the results indicated that only superior sulcus depth can be directly influenced by early intervention of skeletal growth patterns. Specifically, inhibiting maxillary overgrowth in Class II might significantly decrease superior sulcus depth, while promoting maxillary growth in Class III might significantly increase superior sulcus depth. In terms of other upper lip characteristics, plenty of confounding factors should be taken into consideration instead of merely modifying skeletal growth. We added the explanation in the revision (Line 383-387).

In closing, we would like to thank the editor and the reviewers once again for the instructive suggestions, and we look forward to hearing from you regarding our submission. We would be glad to respond to any further questions and comments that you may have.

Reference:

Gauss, T., Ageron, F. X., Devaud, M. L., Debaty, G., Travers, S., Garrigue, D., Raux, M., Harrois, A., Bouzat, P., & French Trauma Research, I. (2019). Association of Prehospital Time to In-Hospital Trauma Mortality in a Physician-Staffed Emergency Medicine System. JAMA Surg, 154(12), 1117-1124. https://doi.org/10.1001/jamasurg.2019.3475

Pincus, D., Ravi, B., Wasserstein, D., Huang, A., Paterson, J. M., Nathens, A. B., Kreder, H. J., Jenkinson, R. J., & Wodchis, W. P. (2017). Association Between Wait Time and 30-Day Mortality in Adults Undergoing Hip Fracture Surgery. JAMA, 318(20), 1994-2003. https://doi.org/10.1001/jama.2017.17606

Shindo, Y., Ito, R., Kobayashi, D., Ando, M., Ichikawa, M., Goto, Y., Fukui, Y., Iwaki, M., Okumura, J., Yamaguchi, I., Yagi, T., Tanikawa, Y., Sugino, Y., Shindoh, J., Ogasawara, T., Nomura, F., Saka, H., Yamamoto, M., Taniguchi, H., Suzuki, R., Saito, H., Kawamura, T., Hasegawa, Y., & Central Japan Lung Study, G. (2015). Risk factors for 30-day mortality in patients with pneumonia who receive appropriate initial antibiotics: an observational cohort study. Lancet Infect Dis, 15(9), 1055-1065. https://doi.org/10.1016/S1473-3099(15)00151-6

Round 2

Reviewer 2 Report

I red your revised manuscript. It can be accepted now.